# Do Green Buildings Have Superior Performance over Non-Certified Buildings? Occupants' Perceptions of Strengths and Weaknesses in Office Buildings

**Maryam Khoshbakht** [1,*] , **Eziaku Rasheed** [2] **and George Baird** [3]

1  Cities Research Institute, Griffith University, Gold Coast 4215, Australia
2  School of Built Environment, Massey University, Auckland 0632, New Zealand
3  School of Architecture, Victoria University of Wellington, Wellington 6011, New Zealand
*  Correspondence: m.kh@griffithuni.edu.au

**Abstract:** The main objectives of green buildings are to improve their design and operation. Many studies have investigated whether green buildings lead to higher occupant satisfaction, yet with contradictory conclusions. The paper builds on the results of post-occupancy evaluation surveys of 68 buildings using the BUS Methodology. Satisfaction scores expressed by occupants with the qualities of their indoor environment were compared between the green and non-certified buildings. This research investigates whether green buildings have superior performance to non-certified buildings from the occupants' perspectives. It was found that generally occupants were more satisfied in green buildings than in non-certified buildings. However, the differences were not significant for any of the environmental and operational parameters including thermal comfort, lighting, noise, and air quality. In the case of operational parameters such as design, needs, image of the building, and cleaning, the differences between the two building groups were notable. Air quality, design and work requirement had the strongest influence on perceived comfort in both green and non-certified buildings. Noise had the strongest influence on perceived productivity for both building groups. Although overall green buildings performed better than the non-certified buildings, the differences between the two were negligible particularly for environmental parameters. Most of the green buildings were not performing entirely as their designers may have intended and had weaknesses that needed to be addressed.

**Keywords:** green buildings; post-occupancy evaluation; building performance; non-certified buildings; sustainable buildings

## 1. Introduction

Green buildings are those structures that are environmentally responsible and resource efficient throughout their life cycle. Green buildings are categorised in this paper as buildings that had certifications from green building organisations such as Green Globe, Green Star, BRE, LEED, Green Design, etc. [1], as opposed to non-certified buildings that are regarded as those without any green building certifications. As private, market-based regulatory mechanisms, green building certifications are a voluntary form of governance [2] that are intended to ensure that buildings are designed, constructed and operated efficiently and reduce or eliminate the negative impact of buildings on occupants and the environment. These green building certifications incentivise reductions in energy, water, and building materials consumption, and the enhancing of occupant health and overall community connectivity [3] through the award of credits or points to buildings. They define green buildings based on environmental, social, and economic sustainability aspects. As such, a green building is expected to achieve an acceptable number of credits as specified by the certification tool and evidence good–best practice in the design of buildings.

A significant body of research has shown green buildings' environmental, social, and economic benefits [4–6]. For example, Baird, Leaman [7] found that self-reported productivity in green buildings was higher than in non-certified buildings because of the Indoor Environment Quality (IEQ)—lighting, temperature, noise and air quality. In two other studies, indoor air quality (IAQ) was rated better in green buildings when compared with non-certified buildings [8,9]. Some studies indicated higher satisfactions with lighting in green buildings [7,10]. Licina and Langer [11] found statistically significant improvement in IAQ satisfaction after occupants moved from non-certified buildings to WELL certified buildings. Prakash (2005) observed that occupants of LEED-certified buildings felt that daylighting and thermal comfort positively affected the occupants' perception of productivity. In their study, Fowler, Rauch [4] noted that green buildings use less energy and water, cost less to maintain, and have more satisfied occupants compared to non-certified buildings.

That said, some other research contradicts the superiority of green buildings over non-certified buildings. Studies report that some green buildings may not provide more comfortable environments for their occupants [12]. A few studies indicated lower satisfaction with indoor air quality in green buildings compared to their non-certified counterparts [13,14]. Indoor environmental controls did not score significantly different in the two groups in another study [7]. A few reports detected no significant differences in the lighting performance of green buildings [13,15]. A few other articles reported lower satisfactions in green buildings with regard to lighting [9,16]. Tham, Wargocki [17] showed that sick leave days were not significantly different between occupants in a Green Mark Platinum-certified building as compared to a non-Green Mark building.

Altomonte et al. indicated that BREEAM certification did not seem to substantively influence building and workspace satisfaction [18]. Another study emphasised that the achievement of a specific IEQ credit did not significantly increase satisfaction with the corresponding IEQ factor [19]. One study showed that LEED buildings tend to perform slightly better in air quality, and slightly worse in terms of the availability of light, but the difference in mean satisfaction scores was negligible [20]. Yet another study indicated that the occupant satisfaction with the main aspects of building in both green and non-certified buildings is lower than the benchmark data [21], although green building occupants tended to be more tolerant of their ambient environments [13]. One study by Rashid et al. [22] found no evidence for direct effects of environmental design features on occupants' environmental awareness and organisational image based on frequency, correlational, and regression analysis. Abbaszadeh et al. [15] emphasised the drawbacks in controls of environmental parameters in open-plan green buildings.

While the contention exists, the concept of green buildings remains foundational to the aspirations of climate responsiveness, energy efficiency and healthy environments. The problem lies with the eventual output of green inspired building designs that show a "performance gap" between the design and performance. Wu et al. [23] observed that evidence is growing that the actual performance of green buildings is not as good as expected. There seems to be a significant difference between the green building performance predictions and measurements at the post-occupancy stage.

While most studies either support or refute the performance of green buildings, few works have explored their strengths and weaknesses from the occupants' point of view [7,8,14]. Moreover, there is a gap in literature about the parameters that most influence perceived comfort and productivity in green and non-certified buildings. Our study fills this gap by providing a detailed and comprehensive study to reveal whether green buildings have a superior operational and environmental performance to their non-certified counterparts. We identify the strengths and weaknesses and highlight where improvements might be sought in green buildings.

## 2. Background

Post-occupancy evaluation (POE) studies are the systematic process of analysing the performance of a building after its occupation. The knowledge acquired from POE studies can produce more accurate performance prediction models and narrow the performance gap in the building industry. POE studies also enable the reliable identification of building performance shortcomings and, therefore, form the basis for improvement measures [24]. Khoshbakht, Gou [25] maintained that the lack of performance evaluations at the post-occupancy stage could further hold back future developments and further widen the performance gap. A satisfaction survey such as a POE study is one of the best practical ways to identify obstacles and errors in building operations while emphasising the needs and values of building occupants [26]. In satisfaction surveys, building occupants are asked to rate their satisfaction with several building performance parameters [27].

The application of satisfaction surveys to POE studies does have its limitations. For example, perception-based evaluations are marred by various biases and accuracy-based errors [28], and lack precision in revealing the building's actual performance [29]. However, it is suggested that including factors such as occupant demographics, background, and building types ensures consistent judgements and can improve satisfaction surveys' reliability [30].

Our study analyses the POEs of 68 buildings globally comprising green and non-certified buildings. In the building sample selection, a few criteria were advanced to keep the dataset consistent and make sure results are representative. The first criterion was that the selected building for study should have had offices for staff that spend most of their time indoors. The second criterion was the building type, making sure the selected building was either commercial office building or university building. We also made sure we could collect responses from at least 75% of the occupants and that the buildings were less than 50 years old. Our sample of green buildings included various global green building certifications including Green Globe, Green Star, BRE, LEED, and Green Design.

All the buildings had both cellular offices and open-plan areas shared by various occupants. The buildings include commercial and academic offices accommodating meeting spaces, seminar rooms, and offices for staff. All green buildings feature a mixed-mode ventilation system. Mixed-mode buildings are those which use a combination of air-conditioning and natural ventilation through windows and air vents. The control of mixed-mode ventilations and switching between air-conditioning and natural ventilation differ among our sample of green buildings. In some green buildings, the control of the mixed-mode ventilation system is fully automated with no opportunity for occupants to override the automated operation of windows (see Figure 1a). In this type, the ventilation system switches between air-conditioning and natural ventilation only if the outdoor temperature falls within the desired temperature range, and occupants have no control over the ventilation operation modes. In type 2 green buildings, occupants have the opportunity to over-ride the automated window operations and manually switch ventilation modes between air-conditioning and natural ventilation (see Figure 1b). In the rest of the building sample (type 3 green buildings), the decision over the mode of ventilation operation is made by occupants, and the automation system only prevents the concurrent operation of air-conditioning and natural ventilation by switching off the air-conditioning when windows are opened (see Figure 1c). In non-certified buildings, ventilation modes were either or both air-conditioning and natural ventilation, and occupants had full control of air-conditioning and operation of windows, the difference from green buildings being that there was no automation system to prevent the concurrent operation of the two ventilation modes, which may result in the loss of conditioned air through open windows and increasing building energy loads.

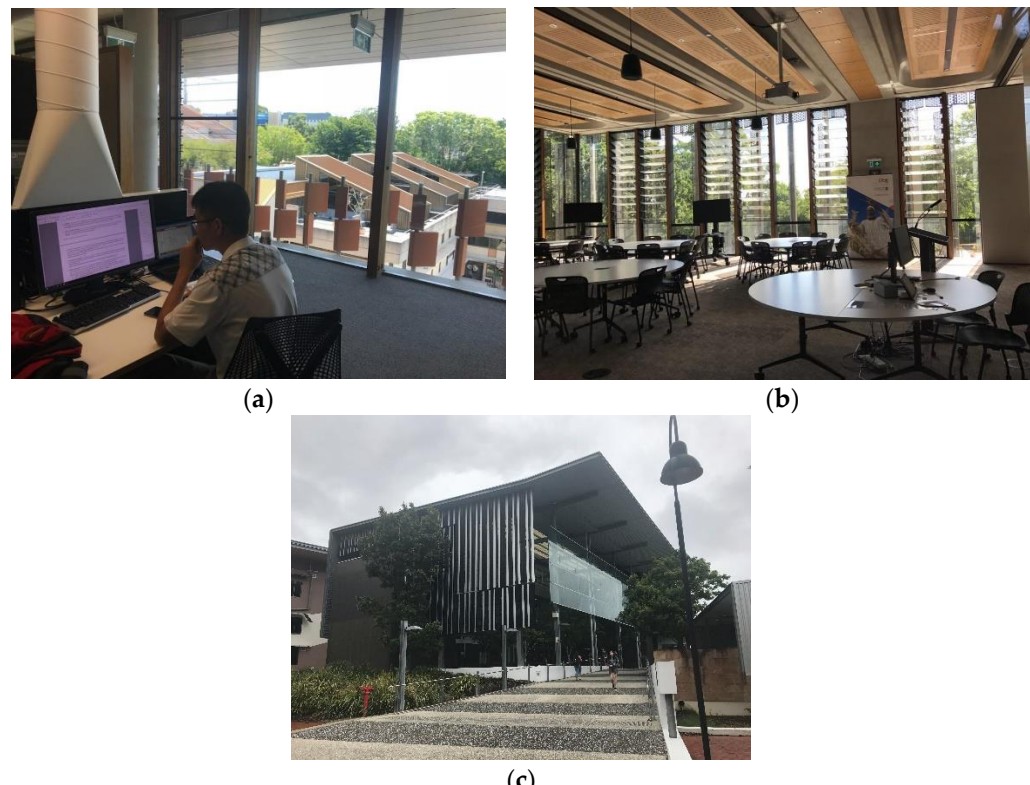

**Figure 1.** Three example buildings in the green building group sample. (**a**–**c**) are fully automated, semi-automated, and manual mixed-mode buildings, respectively.

Our study offers a systematic comparison of the performance of green and non-certified buildings from the occupants' perspective and identifies possible strengths and weaknesses. This study also identifies the parameters that most influence perceived comfort and productivity in green and non-certified buildings.

## 3. Methodology

### 3.1. POE Surveys Protocol

The BUS Methodology questionnaire was used for the occupant satisfaction surveys in this study. The BUS Methodology survey has standard questionnaire templates that have been utilised in numerous research projects worldwide [31]. Acting as a tool for optimising operational buildings, the BUS questions were designed to identify features that work well and improve buildings from the occupants' perspective [32]. The questions cover two major areas:

- *Background information*: this includes basic participant information regarding age, gender, the location of participant desks in terms of distance to a window, and the size of workgroups;
- *Satisfaction score*: this includes questions about environmental, operational, overall comfort, productivity, and control.

These areas provided robust data about the demographics of building occupants and their perceptions. It enabled us to understand the reasons for the satisfaction scores by relating them to their background information. The BUS Methodology survey consists of questions covering environmental, operational and satisfaction parameters [7]. Environmental parameters include overall satisfaction with temperature and air, noise, and lighting; operational parameters seek occupants' perceptions regarding building image, design, space, safety, cleaning, the availability of meeting rooms and storage.

The question related to needs asks occupants how facilities in the building meet their needs. Spaces in buildings asks about the efficiency of the use of spaces. The question about

image asks how participants rate the image of the building as a whole presented to visitors. Regarding cleaning and meeting rooms, occupants are asked to rate their satisfaction with the cleaning and the availability of meeting rooms, respectively. For storage, occupants were asked to rate their satisfaction with suitability of storage arrangements and for work requirements, how facilities meet the requirements for their specific work. For furniture, occupants were asked to rate the usability of furniture available at their desks and work areas. Space at desks, specifically asked whether occupants have enough space at their desks and work areas.

Personal control questions address occupants' perception regarding personal control over heating, cooling, ventilation, lighting and noise. Two further questions ask about perception regarding occupants' overall comfort, and productivity. Further details of the BUS Methodology and the questionnaire are described in earlier research [7,28]. The responses were sought on a 7-point scale for all parameters except for productivity, which was based on a 9-point scale (see Table 1).

**Table 1.** Questions used for the POE study.

| Parameters | Questions | Responses | |
|---|---|---|---|
| | | Score = 1 | Score = 7 |
| Environmental | Temperature Overall in summer | Uncomfortable | Comfortable |
| | Temperature Overall in winter | Too hot | Too cold B |
| | Air overall in summer | Stable | Varies during the day |
| | Air overall in winter | Unsatisfactory | Satisfactory |
| | Natural light | Too little | Too much B |
| | Artificial light | Too little | Too much B |
| | Noise From colleagues | Too little | Too much B |
| | Noise From outside | Too little | Too much B |
| Operational | Design | Unsatisfactory | Satisfactory |
| | Needs | Unsatisfactory | Satisfactory |
| | Space in building | Ineffective | Effective |
| | Image to visitors | Poor | Good |
| | Cleaning | Unsatisfactory | Satisfactory |
| | Meeting rooms availability | Unsatisfactory | Satisfactory |
| | Storage arrangements suitability | Unsatisfactory | Satisfactory |
| | Facilities meet work requirements | Very poorly | Very well |
| | Furniture | Very poorly | Very good |
| | Space at Desk | Very poorly | Very good |
| Personal Control | Control Heating | No control | Full control |
| | Control Cooling | No control | Full control |
| | Control Ventilation | No control | Full control |
| | Control Lighting | No control | Full control |
| | Control Noise | No control | Full control |
| Overall Comfort | Overall comfort | Unsatisfactory | Satisfactory |
| Productivity | Productivity | Productivity decreased by 40% or less (Score = 1) | Productivity increased by 40% or more (Score = 9) |

B indicates bipolar scales—all the others are unipolar.

To obtain a minimum 75% response rate from each building, the questionnaires were both distributed and collected in person by the authors. The completed questionnaires were collected either on the day or not later than a week after the distribution. A total of 5098 completed questionnaires were collected from 68 buildings.

### 3.2. Dataset Characteristics

A summary of building specifications in the dataset is presented in Table 2. As indicated in Table 2, just over half the buildings in our dataset were located in New Zealand (39 buildings), while others were in a range of countries, including Australia, England, USA, India, Ireland, Japan, Malaysia, and Canada. The average number of participants from each building in our dataset was 75. In terms of building use, our dataset consists of commercial buildings (72%), and academic buildings (28%). Commercial buildings are used primarily for business purposes, and academic buildings are used for university teaching, research and administration. For ventilation, 43% of buildings in our dataset had natural ventilation, 31% had air-conditioning, and 26% had a combination of natural ventilation and air-conditioning (mixed-mode) systems.

**Table 2.** Building specifications of the studied buildings.

| Variables | Dataset Distribution |
|---|---|
| Building size (number of participants) | Smallest building had 11 participants; Largest building had 342 participants; By average buildings in our dataset had around 75 participants; |
| Percentage rate of occupant participation | 75% or more |
| Design intent | Green buildings (38 buildings); Non-certified buildings (30 buildings) |
| Country | New Zealand (39); Australia (12); England (8); USA (1); India (2); Ireland (1); Japan (1); Malaysia (2); Canada (2) |
| Building use | Commercial (72%); Academic (28%) |
| Ventilation | Natural ventilation (43%); Air-conditioning (31%); Mixed-mode (26%) |

For both sets of buildings the majority of participants were 30 years or over (69% in green and 81% in non-certified buildings). In green buildings the percentage rate of female participants was 54% and male participants was 46%, while in non-certified buildings the percentage rate of female participants was 47% and male participants was 53%. For both building types, almost half of the participants were seating next to a window (53% in green and 49% in non-certified buildings). *Workgroups* consist of five categories: *solo* with a single occupant; *duo* with two occupants; *3–4* with three or four occupants; *5–9* with five to nine occupants; and *over 9* with more than nine occupants. In terms of space sharing with others, for most participants in both building groups the room was shared with *over 9* people. More details of space sharing in comparing the two building types are provided in Table 3 for comparison purposes.

### 3.3. Data Analysis

Independent sample T-tests were adopted for comparing the two groups of green and non-certified buildings in terms of environmental parameters, operational parameters, control parameters, and overall comfort and productivity scores. The standardized size of the mean difference (effect size index) between the two building groups, green and non-certified, was calculated using one-way multivariate analysis of variance (MANOVA). The effect size (Rho) is an index for calculating the magnitude of effect or association between two or more predictor variables [20]. An effect size smaller than 0.20 is classed as negligible, between 0.20 and 0.50 as small, between 0.50 and 0.8 as medium, and larger than 0.8 as large effect sizes according to Cohen [33]. Effect sizes are the difference between the two groups divided by the standard deviation of one of the groups. Effect sizes in the analysis of occupants' perceived satisfaction has been used in a considerable number of papers in the past such as Candido, Marzban [34], who compared occupants' satisfaction, perceived productivity and health between two groups of certified premises against other open-plan offices. For example, Marín-Restrepo, Trebilcock [35] also compared effect sizes and used Cohen's d to identify patterns in the occupants' adaptive behaviours and spatial layouts in office environments.

**Table 3.** The distribution of individual occupant responses in terms of personal and spatial factors.

| | Participants' Categories | Green Buildings | | Non-Certified Buildings | | Total Number |
|---|---|---|---|---|---|---|
| Age | Under 30 | 843 | 31% | 446 | 19% | 1289 |
| | 30 or above | 1884 | 69% | 1842 | 81% | 3726 |
| Gender | Female | 1452 | 54% | 1075 | 47% | 2527 |
| | Male | 1262 | 46% | 1209 | 53% | 2471 |
| Window | Sit next to a window | 1440 | 53% | 1135 | 49% | 2575 |
| | No window nearby | 1293 | 47% | 1164 | 51% | 2457 |
| Work groups | Solo (occupied only by one) | 687 | 26% | 440 | 19% | 1127 |
| | Duo (occupied by two) | 215 | 8% | 180 | 8% | 395 |
| | 3–4 (occupied by 3–4 people) | 406 | 15% | 420 | 19% | 826 |
| | 5–9 (occupied by 5–9 people) | 393 | 15% | 346 | 15% | 739 |
| | Over 9 (occupied by more than 9 people) | 974 | 36% | 881 | 39% | 1855 |

## 4. Results

### 4.1. Environmental Parameters

The green buildings were compared to their non-certified counterparts for eight environmental parameters using the T-test and mean ranks (Table 4). $\Delta M$ is the median score of green buildings minus the median score of non-certified buildings. The $\Delta M$ scores were also plotted and shown in Figure 2. For most parameters, green buildings achieved a better score except for the noise from colleagues and artificial lighting. Thermal comfort in winter ($\Delta M = +0.26$), thermal comfort in summer ($\Delta M = +0.25$), air quality in winter ($\Delta M = +0.35$), and air quality in summer ($\Delta M = +0.34$) all scored better in green buildings. Noise from outside ($\Delta M = +0.05$) and natural lighting ($\Delta M = +0.32$) were also scored better in green buildings. Noise from colleagues ($\Delta M = -0.20$) and artificial lighting ($\Delta M = -0.22$) were scored worse in green buildings comparatively.

**Table 4.** Mean differences and effect size calculation comparing green and non-certified buildings regarding environmental parameters.

| | Groups | N | Median | Std. Deviation | 95% Confidence Interval | | $\Delta M$ | Cohen's d | Effect Size (Rho) | F | Sig. |
|---|---|---|---|---|---|---|---|---|---|---|---|
| | | | | | Lower Bound | Upper Bound | | | | | |
| Thermal Comfort Winter | 1 | 2492 | 4.73 | 1.71 | 4.66 | 4.79 | 0.26 | 0.15 | 0.08 | 26.81 | <0.001 *** |
| | 2 | 2141 | 4.47 | 1.70 | 4.39 | 4.54 | | | | | |
| Air Winter Overall | 1 | 2507 | 4.69 | 1.63 | 4.62 | 4.75 | 0.35 | 0.22 | 0.11 | 52.38 | <0.001 *** |
| | 2 | 2171 | 4.34 | 1.61 | 4.27 | 4.41 | | | | | |
| Thermal Comfort Summer | 1 | 2167 | 4.54 | 1.79 | 4.46 | 4.62 | 0.25 | 0.14 | 0.07 | 19.84 | <0.001 *** |
| | 2 | 1775 | 4.29 | 1.66 | 4.22 | 4.37 | | | | | |
| Air Summer Overall | 1 | 2160 | 4.56 | 1.68 | 4.49 | 4.63 | 0.34 | 0.21 | 0.10 | 43.23 | <0.001 *** |
| | 2 | 1764 | 4.22 | 1.57 | 4.15 | 4.29 | | | | | |
| Noise Colleagues | 1 | 2644 | 4.34 | 1.35 | 4.29 | 4.39 | −0.20 | −0.14 | −0.07 | 24.64 | <0.001 *** |
| | 2 | 2221 | 4.54 | 1.48 | 4.48 | 4.61 | | | | | |
| Noise Outside | 1 | 2621 | 3.87 | 2.20 | 3.79 | 3.96 | 0.05 | 0.03 | 0.01 | 0.86 | 0.35 |
| | 2 | 2200 | 3.82 | 1.28 | 3.77 | 3.88 | | | | | |
| Natural Lighting | 1 | 2648 | 3.98 | 1.34 | 3.93 | 4.03 | 0.32 | 0.23 | 0.11 | 65.27 | <0.001 *** |
| | 2 | 2220 | 3.66 | 1.46 | 3.59 | 3.72 | | | | | |
| Artificial Lighting | 1 | 2646 | 4.22 | 1.05 | 4.18 | 4.26 | −0.22 | −0.21 | −0.10 | 51.53 | <0.001 *** |
| | 2 | 2215 | 4.44 | 1.06 | 4.39 | 4.48 | | | | | |

Group 1 means green buildings and Group 2 means non-certified buildings. N is the number of buildings. $\Delta M$ is the median scores of Group 1 minus Group 2. F is the F-distribution. Sig. is the significance of the results of the *p*-value. *** *p*-value less than 0.001.

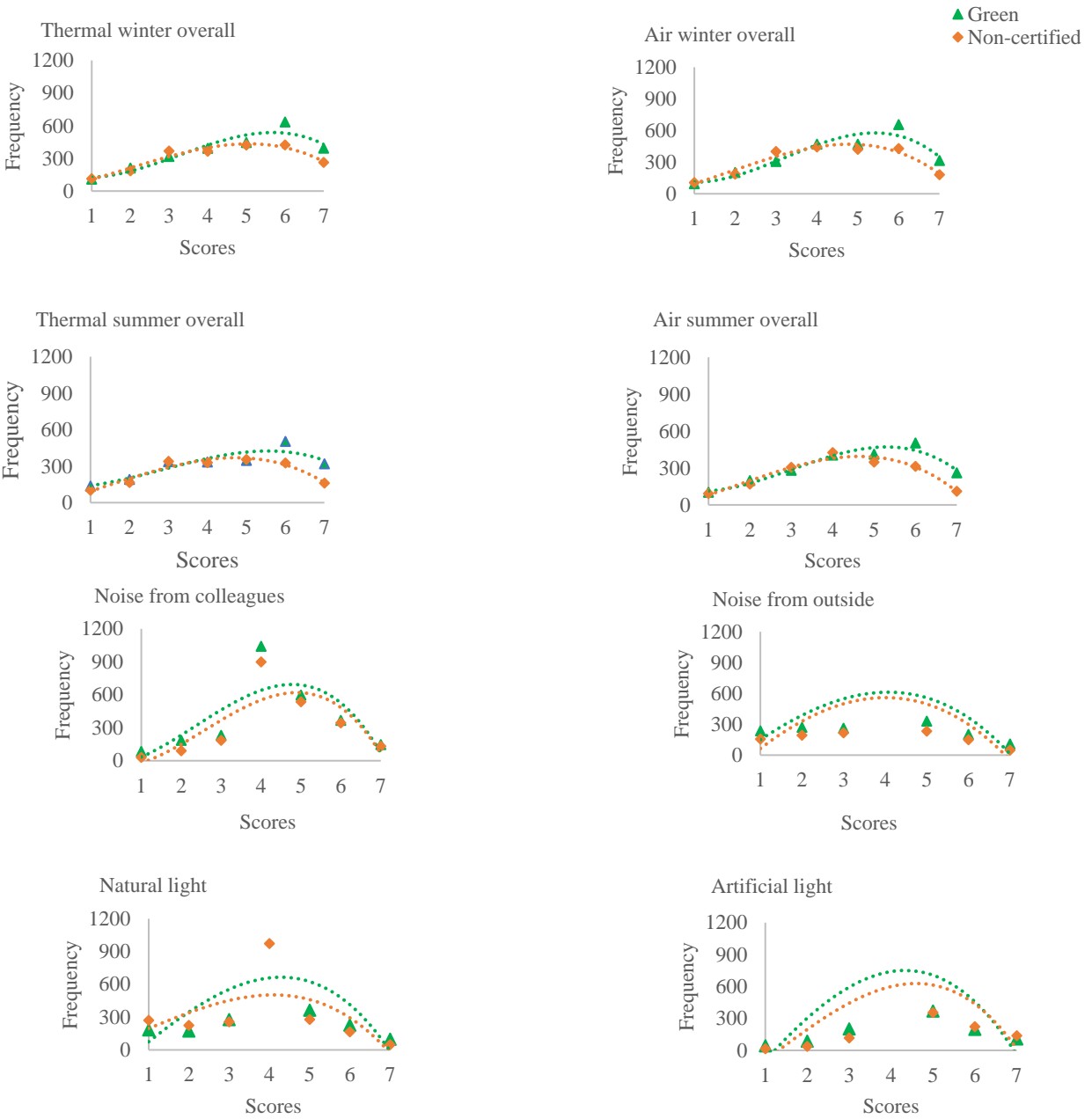

**Figure 2.** Distribution characteristics of median scores for environmental parameters.

The effect size calculation revealed the significance of differences between the performance of green and non-certified buildings. The differences between the green and non-certified buildings were negligible for all environmental parameters according to effect size calculations with all Rho less than 0.20.

This is further confirmed in Figure 2, which illustrates the distribution characteristics of mean scores for environmental parameters. As illustrated in Figure 2, the curves reach their peaks almost in the middle of the 7-point scale values for noise and lighting parameters, while for thermal comfort and air quality, the curves are slightly skewed towards the upper band, indicating that thermal comfort and air quality achieved better scores in comparison to noise and lighting parameters for both green and non-certified buildings.

### 4.2. Operational Parameters

The performance of the green buildings was compared to non-certified buildings for 10 operational parameters using the T-test and mean ranks (Table 5). For all parameters,

the green buildings obtained better scores, indicating that they were outperforming non-certified buildings for all operational parameters, including design ($\Delta M$ = +0.75), needs ($\Delta M$ = +0.60), spaces in buildings ($\Delta M$ = +0.46), image ($\Delta M$ = +1.39), cleaning ($\Delta M$ = +0.84), meeting rooms ($\Delta M$ = +0.65), storage ($\Delta M$ = +0.37), work requirement ($\Delta M$ = +0.41), furniture ($\Delta M$ = +0.23), and space at desk ($\Delta M$ = +0.02). The questions related to design specifically asked participants how satisfied they are about the general design of the building.

**Table 5.** Mean differences and effect size calculation comparing green and non-certified buildings regarding operational parameters.

| | Groups | N | Median | Std. Deviation | 95% Confidence Interval | | $\Delta M$ | Cohen's d | Effect Size (Rho) | F | Sig. |
| | | | | | Lower Bound | Upper Bound | | | | | |
|---|---|---|---|---|---|---|---|---|---|---|---|
| Design | 1 | 2733 | 5.34 | 1.48 | 5.28 | 5.39 | 0.75 | 0.50 | 0.24 | 307.85 | <0.001 *** |
| | 2 | 2278 | 4.59 | 1.50 | 4.53 | 4.66 | | | | | |
| Need | 1 | 2708 | 5.34 | 1.46 | 5.28 | 5.39 | 0.6 | 0.41 | 0.20 | 199.71 | <0.001 *** |
| | 2 | 2262 | 4.74 | 1.50 | 4.68 | 4.80 | | | | | |
| Space in Building | 1 | 2711 | 5.02 | 1.54 | 4.96 | 5.08 | 0.46 | 0.30 | 0.15 | 115.59 | <0.001 *** |
| | 2 | 2264 | 4.56 | 1.49 | 4.49 | 4.62 | | | | | |
| Image | 1 | 2732 | 5.97 | 1.26 | 5.92 | 6.01 | 1.39 | 0.95 | 0.43 | 1132.83 | <0.001 *** |
| | 2 | 2282 | 4.58 | 1.65 | 4.52 | 4.65 | | | | | |
| Cleaning | 1 | 2741 | 5.38 | 1.50 | 5.33 | 5.44 | 0.84 | 0.54 | 0.26 | 366.18 | <0.001 *** |
| | 2 | 2297 | 4.54 | 1.63 | 4.47 | 4.60 | | | | | |
| Meeting | 1 | 2634 | 5.02 | 1.71 | 4.95 | 5.08 | 0.65 | 0.37 | 0.18 | 168.13 | <0.001 *** |
| | 2 | 2260 | 4.37 | 1.77 | 4.30 | 4.45 | | | | | |
| Storage | 1 | 2608 | 4.40 | 1.76 | 4.33 | 4.46 | 0.37 | 0.21 | 0.11 | 52.92 | <0.001 *** |
| | 2 | 2240 | 4.03 | 1.69 | 3.96 | 4.10 | | | | | |
| Work Requirement | 1 | 2654 | 5.32 | 1.36 | 5.27 | 5.38 | 0.41 | 0.29 | 0.14 | 109.34 | <0.001 *** |
| | 2 | 2226 | 4.91 | 1.43 | 4.85 | 4.96 | | | | | |
| Furniture | 1 | 2700 | 5.33 | 1.35 | 5.28 | 5.38 | 0.23 | 0.17 | 0.08 | 34.08 | <0.001 *** |
| | 2 | 2275 | 5.10 | 1.37 | 5.05 | 5.16 | | | | | |
| Space at Desk | 1 | 2698 | 4.34 | 1.41 | 4.29 | 4.39 | 0.02 | 0.01 | 0.006 | 0.13 | 0.72 |
| | 2 | 2278 | 4.32 | 1.72 | 4.25 | 4.39 | | | | | |

Group 1 means green buildings and Group 2 means non-certified buildings. N is the number of buildings. $\Delta M$ is the median scores of Group 1 minus Group 2. F is the F-distribution. Sig. is the significance of the results of the *p*-value. *** *p*-value less than 0.001.

The T-test revealed the differences between the performance of green and non-certified buildings were small for image of the building (Rho = 0.43) and cleaning (Rho = 0.26) and design (Rho = 0.24). For parameters such as needs (Rho = 0.20), the differences were negligible.

As also confirmed in the illustrations in Figure 3, the gaps between the two curves were much more obvious for design, image of the building, and cleaning. This illustration further confirms that green buildings outperform their non-certified counterparts by obtaining moderately better scores for design, image of the building, and cleaning. For green buildings, the curves reach their peaks closer to a score of 7 for all operational parameters, while for non-certified buildings, the peaks are slightly closer to the middle point in the 7-point scale.

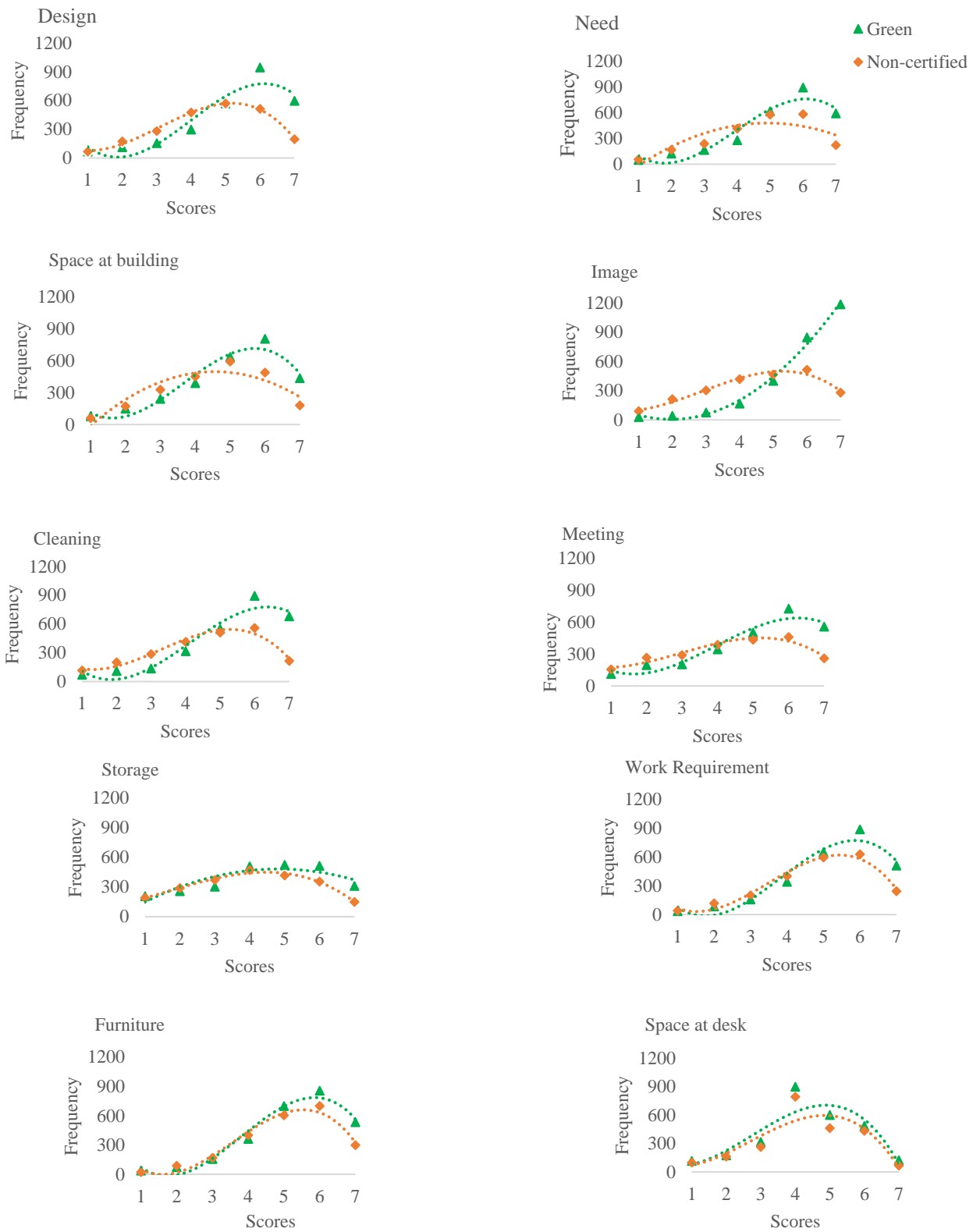

**Figure 3.** Distribution characteristics of median scores for operational parameters.

### 4.3. Perceived Control Parameters

The performance of green buildings was also analysed in comparison to their non-certified counterparts for perceived indoor environmental quality control parameters (Table 6). The null hypothesis test revealed statistically significant differences between the performance of green and non-certified buildings for all five perceived personal control

parameters (*p*-value < 0.001). For all parameters, green buildings obtained a better score, indicating that green buildings outperformed non-certified buildings for all perceived personal control parameters, including heating (ΔM = +0.37), cooling (ΔM = +0.41), ventilation (ΔM = +0.87), lighting (ΔM = +0.69), and noise (ΔM = +0.23). Personal controls asked occupants how much control they had over adjusting heating, cooling, ventilation, lighting and noise in their work areas.

**Table 6.** Mean differences and effect size calculation comparing green and non-certified buildings regarding control parameters.

| | Groups | N | Median | Std. Deviation | 95% Confidence Interval | | ΔM | Cohen's d | Effect Size (Rho) | F | Sig. | Importanc (% yes) |
|---|---|---|---|---|---|---|---|---|---|---|---|---|
| | | | | | Lower Bound | Upper Bound | | | | | | |
| Control Heating | 1 | 2567 | 2.45 | 1.86 | 2.38 | 2.53 | 0.37 | 0.20 | 0.10 | 50.52 | <0.001 *** | >99% |
| | 2 | 2200 | 2.08 | 1.74 | 2.01 | 2.15 | | | | | | |
| Control Cooling | 1 | 2577 | 2.48 | 1.80 | 2.41 | 2.55 | 0.41 | 0.23 | 0.12 | 63.43 | <0.001 *** | >99% |
| | 2 | 2195 | 2.07 | 1.70 | 2.00 | 2.15 | | | | | | |
| Control Ventilation | 1 | 2574 | 2.98 | 2.02 | 2.90 | 3.06 | 0.87 | 0.46 | 0.23 | 254.31 | <0.001 *** | >99% |
| | 2 | 2193 | 2.11 | 1.71 | 2.03 | 2.18 | | | | | | |
| Control Lighting | 1 | 2582 | 3.37 | 2.15 | 3.28 | 3.45 | 0.69 | 0.33 | 0.16 | 127.93 | <0.001 *** | >99% |
| | 2 | 2193 | 2.68 | 2.00 | 2.60 | 2.77 | | | | | | |
| Control Noise | 1 | 2586 | 2.33 | 1.59 | 2.27 | 2.39 | 0.23 | 0.15 | 0.08 | 27.79 | <0.001 *** | >99% |
| | 2 | 2198 | 2.10 | 1.44 | 2.04 | 2.16 | | | | | | |

Group 1 means green buildings and Group 2 means non-certified buildings. N is the number of buildings. ΔM is the median scores of Group 1 minus Group 2. F is the F-distribution. Sig. is the significance of the results of the *p*-value. *** *p*-value less than 0.001.

The T-test calculations also revealed that differences between green and non-certified buildings were negligible for all personal control parameters except for ventilation (Rho = 0.23), which was classed as small. This is further confirmed in the illustration in Figure 4, where the gap between the two curves is the largest for personal control over ventilation with the highest effect size among the five parameters.

In the surveys, respondents were also asked about the importance of personal controls over heating, cooling, ventilation, lighting and noise. The responses were compared between green and non-certified buildings (Table 7). All variables obtained more than 25%, indicating that such variable was important for more than 25% of participants in the survey. Although 25% seems like a small number, more studies of these types of questions are needed to develop a benchmark for analysing the significance of responses. What is evident from the results is that perceived controls over heating achieved the highest score among the five variables for both green and non-certified buildings. This indicates that heating seems to be the most important for participants among the five variables. Comparing the two building types, overall, non-certified buildings obtained relatively higher percentages for all five variables. This may indicate that green buildings performed better in providing personal control over the five environmental variables. More significantly, the difference between the scores for green and non-certified buildings was larger for personal control over heating, cooling and noise.

**Table 7.** "Importance of personal control" question where respondents ticked a box if it was important to them.

| Buildings | Importance of Control Heating | Importance of Control Cooling | Importance of Control Ventilation | Importance of Control Lighting | Importance of Control Noise |
|---|---|---|---|---|---|
| Green | 27% | 26% | 26% | 25% | 25% |
| Non-certified | 31% | 29% | 27% | 25% | 29% |

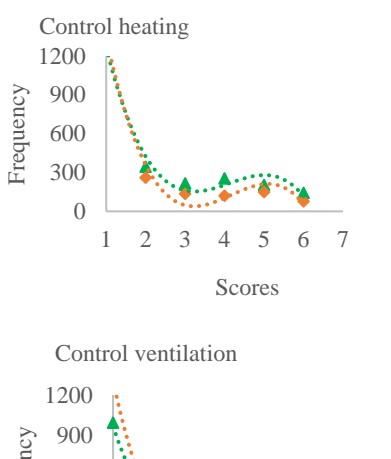
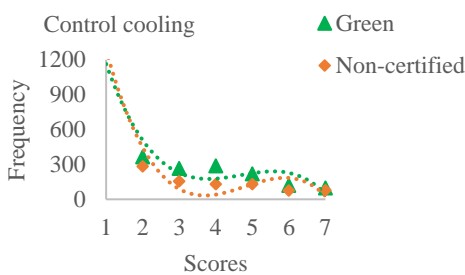
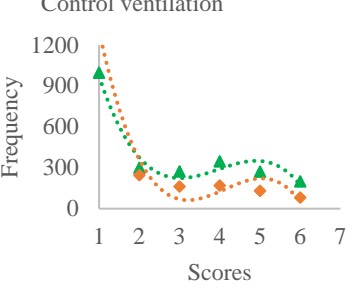
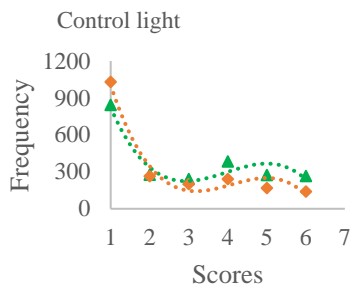
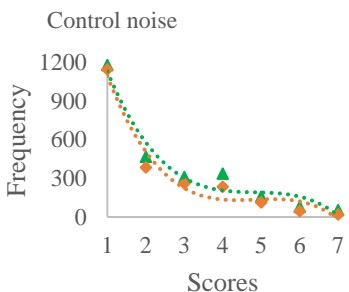

**Figure 4.** Distribution characteristics of median scores for perceived control parameters.

### 4.4. Perceived Comfort and Productivity

Perceived comfort and productivity scores were compared between the green and non-certified buildings using T-tests. For both perceived comfort and perceived productivity, green buildings obtained better scores (Table 8). Perceived comfort was scored better in green buildings with ΔM of +0.56. Perceived productivity was also scored better in green buildings with ΔM of +0.78.

**Table 8.** Comparison of mean values of overall comfort and productivity scores between green and non-certified buildings.

| | Groups | N | Median | Std. Deviation | 95% Confidence Interval | | ΔM | Cohen's d | Effect Size (Rho) | F | Sig. |
|---|---|---|---|---|---|---|---|---|---|---|---|
| | | | | | Lower Bound | Upper Bound | | | | | |
| Perceived comfort | 1 | 2663 | 5.12 | 1.44 | 5.06 | 5.17 | 0.56 | 0.39 | 0.19 | 178.11 | <0.001 *** |
| | 2 | 2220 | 4.56 | 1.45 | 4.50 | 4.62 | | | | | |
| Perceived productivity | 1 | 2541 | 5.49 | 1.70 | 5.42 | 5.56 | 0.78 | 0.48 | 0.23 | 260.59 | <0.001 *** |
| | 2 | 2135 | 4.71 | 1.58 | 4.64 | 4.78 | | | | | |

Group 1 means green buildings and Group 2 means non-certified buildings. N is the number of buildings. ΔM is the difference between the median scores of Group 1 minus Group 2. F is the F-distribution. Sig. is the significance of the results of the *p*-value. *** *p*-value less than 0.001.

The T-test analysis revealed that the differences between green and non-certified buildings were small for perceived productivity (Rho = 0.23). The difference in the perfor-

mance of green and non-certified buildings in perceived overall comfort was negligible (Rho = 0.19).

Ordinal regression was used to analyse which parameters influence perceived comfort and productivity. Perceived comfort was regressed with 8 environmental, 10 operational and 5 control parameters, once for green and once for non-certified buildings (Table 9). A total of 12 factors showed significance in regressing with perceived comfort for green buildings. They include thermal comfort in winter, air quality in winter, air quality in summer, noise from colleagues, natural lighting design, needs, space in buildings, image, work requirement, furniture and control over noise. Demographics including age and gender did not significantly influence perceived comfort for either green or non-certified buildings.

**Table 9.** Ordinal regression of perceived comfort comparing green and non-certified buildings.

| Covariates | Estimate | | Std. Error | | Sig. | |
|---|---|---|---|---|---|---|
| | **Green** | **Non-Certified** | **Green** | **Non-Certified** | **Green** | **Non-Certified** |
| Temperature Winter Overall | 0.20 | 0.14 | 0.05 | 0.05 | <0.001 *** | <0.05 * |
| Air Winter Overall | 0.33 | 0.44 | 0.06 | 0.06 | <0.001 *** | <0.001 *** |
| Temperature Summer Overall | 0.06 | 0.11 | 0.05 | 0.05 | 0.21 | <0.05 * |
| Air Summer Overall | 0.41 | 0.36 | 0.06 | 0.06 | <0.001 *** | <0.001 *** |
| Noise from Colleagues | −0.09 | −0.08 | 0.04 | 0.03 | <0.05 * | <0.05 * |
| Noise from Outside | −0.03 | 0.02 | 0.04 | 0.04 | 0.50 | 0.71 |
| Natural Light | 0.09 | 0.19 | 0.04 | 0.04 | <0.05 * | <0.001 *** |
| Artificial Light | 0.00 | −0.08 | 0.05 | 0.05 | 0.99 | 0.11 |
| Design | 0.32 | 0.36 | 0.06 | 0.05 | <0.001 *** | <0.001 *** |
| Need | 0.22 | 0.09 | 0.05 | 0.05 | <0.001 *** | <0.05 * |
| Space at building | 0.11 | 0.04 | 0.04 | 0.05 | <0.05 * | 0.39 ** |
| Image | 0.21 | 0.08 | 0.05 | 0.04 | <0.001 *** | 0.06 |
| Cleaning | 0.02 | 0.05 | 0.04 | 0.04 | 0.64 | 0.15 |
| Meeting | 0.06 | 0.05 | 0.03 | 0.03 | 0.10 | 0.12 |
| Storage | −0.05 | 0.05 | 0.04 | 0.04 | 0.22 | 0.17 |
| Work Requirement | 0.37 | 0.46 | 0.05 | 0.05 | <0.001 *** | <0.001 *** |
| Furniture | 0.18 | 0.18 | 0.05 | 0.05 | <0.001 *** | <0.001 *** |
| Space at Desk | −0.03 | 0.01 | 0.04 | 0.04 | 0.44 | 0.89 |
| Control Heat | −0.03 | 0.00 | 0.04 | 0.06 | 0.44 | 0.95 |
| Control Cool | −0.02 | 0.07 | 0.05 | 0.06 | 0.62 | 0.24 |
| Control Vent | 0.01 | −0.05 | 0.03 | 0.04 | 0.74 | 0.25 |
| Control Light | 0.03 | 0.03 | 0.03 | 0.03 | 0.27 | 0.40 |
| Control Noise | 0.09 | 0.08 | 0.04 | 0.04 | <0.05 * | 0.05 |
| Age | 0.06 | 0.26 | 0.11 | 0.10 | 0.54 | <0.05 * |
| Gender | 0.20 | 0.00 | 0.10 | 0.10 | 0.84 | 0.84 |

Sig. is the significance of the results of the *p*-value. * *p*-value less than 0.05; ** *p*-value less than 0.01; and *** *p*-value less than 0.001.

A slightly different 12e factors also showed significance in regressing with perceived comfort for non-certified buildings. They include thermal comfort in winter, air quality in

winter, thermal comfort in summer, noise from colleagues, natural lighting, design, needs, air in summer, noise from colleagues, design, needs, space in buildings, work requirement, furniture, natural lighting and control over noise. For demographics including age and gender, only age significantly influenced perceived comfort for both green and non-certified buildings. Comparing the coefficients in the model revealed that for green and non-certified buildings, perceived comfort was regressed with the higher value for air quality, design and work requirement.

Perceived productivity was regressed with 8 environmental, 10 operational and 5 control parameters once for green and once for non-certified buildings (Table 10). For green buildings, air quality in winter, air quality in summer, noise from colleagues, noise from outside, natural lighting, design, space in buildings, storage, work requirement, space at desks and control over heating and control over noise showed significance in regressing with perceived productivity. Demographics including age and gender did not significantly influence perceived comfort for both green and non-certified buildings.

**Table 10.** Ordinal regression of perceived productivity comparing green and non-certified buildings.

| Covariates | Estimate | | Std. Error | | Sig. | |
|---|---|---|---|---|---|---|
| | **Green** | **Non-Certified** | **Green** | **Non-Certified** | **Green** | **Non-Certified** |
| Temperature Winter Overall | 0.07 | −0.05 | 0.05 | 0.05 | 0.15 | 0.36 |
| Air Winter Overall | 0.14 | 0.18 | 0.05 | 0.06 | <0.05 * | <0.001 *** |
| Temperature Summer Overall | −0.01 | 0.09 | 0.04 | 0.05 | 0.76 | 0.06 |
| Air Summer Overall | 0.18 | 0.12 | 0.05 | 0.05 | <0.001 *** | <0.05 * |
| Noise from Colleagues | −0.23 | −0.19 | 0.04 | 0.03 | <0.001 *** | <0.001 *** |
| Noise from Outside | −0.06 | −0.04 | 0.02 | 0.04 | <0.001 *** | 0.28 |
| Natural Light | 0.16 | 0.15 | 0.04 | 0.04 | <0.001 *** | <0.001 *** |
| Artificial Light | 0.01 | −0.06 | 0.05 | 0.05 | 0.82 | 0.25 |
| Design | 0.22 | 0.14 | 0.05 | 0.05 | <0.001 *** | <0.001 *** |
| Need | 0.06 | 0.11 | 0.05 | 0.05 | 0.25 | <0.05 * |
| Space at building | 0.09 | −0.04 | 0.04 | 0.04 | <0.05 * | 0.33 |
| Image | 0.05 | 0.06 | 0.05 | 0.04 | 0.29 | 0.13 |
| Cleaning | 0.04 | −0.02 | 0.03 | 0.03 | 0.18 | 0.54 |
| Meeting | 0.04 | 0.04 | 0.03 | 0.03 | 0.21 | 0.27 |
| Storage | −0.08 | 0.06 | 0.04 | 0.04 | <0.05 * | 0.09 |
| Work Requirement | 0.28 | 0.28 | 0.05 | 0.05 | <0.001 *** | <0.001 *** |
| Furniture | −0.06 | −0.11 | 0.04 | 0.05 | 0.21 | <0.05 * |
| Space at Desk | 0.13 | 0.16 | 0.04 | 0.04 | <0.001 *** | <0.001 *** |
| Control Heat | 0.10 | −0.01 | 0.04 | 0.06 | <0.05 * | 0.93 |
| Control Cool | 0.00 | 0.09 | 0.04 | 0.06 | 0.95 | 0.11 |
| Control Vent | 0.04 | −0.03 | 0.03 | 0.04 | 0.21 | 0.51 |
| Control Light | 0.02 | 0.04 | 0.03 | 0.03 | 0.43 | 0.25 |
| Control Noise | 0.08 | 0.04 | 0.04 | 0.04 | <0.05 * | 0.29 |
| Age | 0.06 | 0.26 | 0.11 | 0.10 | 0.54 | <0.05 * |
| Gender | 0.02 | 0.00 | 0.10 | 0.10 | 0.84 | 0.84 |

* $p$-value less than 0.05; and *** $p$-value less than 0.001.

For non-certified buildings, air quality in winter, air quality in summer, noise from colleagues, natural lighting, design, needs, work requirement, furniture, and space at desks showed significance in regressing with perceived productivity. For demographics including age and gender, only age significantly influenced perceived comfort for both green and non-certified buildings. Comparing the coefficients in the model revealed that for both

groups, perceived productivity was highly regressed with noise from colleagues and work requirements for both building groups.

## 5. Discussion

The findings of this study show that the 38 green buildings in our dataset of 68 buildings performed better than their non-certified counterparts for most of the environmental and operational parameters examined. However, the differences were negligible for all environmental parameters and only small for three out of ten operational parameters including design, image of the building, and cleaning. For personal controls, the differences were only small for ventilation. The differences were also negligible for perceived overall comfort and only small for perceived productivity. It is generally believed that occupants perceive green buildings more favourably [36–40]. While green buildings are expected to do better, our results showed that the differences were frequently negligible. Considering the higher construction costs of green buildings, this global review of green buildings' performance showed that certification systems do not necessarily ensure superior performances for green buildings as predicted by designers.

There were even some parameters where green buildings scored lower than non-certified buildings such as artificial lighting and noise from colleagues. The occupants of green buildings were also less satisfied with the artificial lighting in the offices. Green buildings often have greater emphasis on natural lighting design and artificial lighting may attract less attention and manipulation by designers. The other reason could be that green buildings have comparatively more shared spaces and open-plan layouts than non-certified buildings and lighting controls need to be a group decision, which may cause an issue with if disagreements arise [41]. Adjustable task lighting could be a solution for increasing satisfactions with artificial lighting particularly in shared spaces, which also positively impacts occupant comfort, and eye fatigue [42]. There is also a potential for adjustable task lighting to reduce electricity consumption in buildings, while increasing occupant comfort [43].

The findings of this research clearly show that people in green buildings reported higher levels of dissatisfaction with noise from colleagues. A plausible reason is that the majority (66%) of the occupants in our dataset shared their workspaces with three or more people. Shared office spaces are more common in modern workspaces, whether they are green or non-certified buildings, which could be a reason for higher dissatisfaction with ambient noise and artificial lighting in such buildings. Noise could be a big issue in shared work areas and past research shows that distractions and interruptions are prevalent in these spaces and significantly impact occupant comfort and productivity [31,44,45]. For example, Rasheed, Khoshbakht [31] found that occupants who shared their workspaces with more than eight people were less satisfied and comfortable with the IEQ in their offices. Past studies have also shown that workers may feel distracted in open-plan offices because of their acoustic design, hindering their creativity [46]. Unlike cellular offices, occupants have less control over the noise that may interrupt their concentration [31]. One way to avoid noise-related stress and distractions is to increase acoustic privacy in office environments, as Haynes and Price [47] proposed. Although total silence is not necessary for office environments, it is essential to provide spaces for concentration and focus while considering personal preferences [48].

While most of the POE research in this field focuses on environmental parameters [49], our study explored operational parameters as well. We found that green buildings were perceived to perform better than non-certified buildings for all the operational parameters, yet the difference was in the moderate range. For design, image of the building, and cleaning, green buildings performed moderately better than non-certified buildings, but for the rest of the parameters the differences were negligible.

It is worth mentioning three areas identified as needing improvement in both green and non-certified buildings in our dataset, i.e., sufficient storage, space at desks, and the provision of adequate control. We discovered that these parameters were perceived to

perform poorest amongst operational parameters (see Table 5). Storage and desk space are useful for the ergonomic comfort of office users. While sufficient storage allows users to store and maintain important documents, a defined desk space enables focus and attention to the task. The ergonomic importance of these parameters relates to occupant comfort and productivity in the workplace. For instance, in one study [50] it was observed that the type of workstation related to occupants' physical activity and stress levels. They noted that workers in open bench seating were more active than those in cubicles. Providing employees with sit-stand desks positively impacts employees' physical well-being, as suggested by a report [51]. Adjustability and storage may also contribute to higher satisfaction and work performance in office buildings [52].

Our study also found that most of the occupants in our dataset had little or no control over the environmental parameters in their workspaces, as their mean score was less than 3.5 (see Table 6). The occupants also noted the importance of having personal control over heating, cooling, ventilation, lighting and noise. We suggest that the lack of control is a plausible cause because most of the workgroups in our dataset are situated in shared or open-plan space wherein they shared their workspaces with more than two people. A common measure adopted in open-plan work spaces to optimise IEQ performance is the automation of environmental control, thus reducing individual occupants' control. This means that occupants have limited controls to tune their environments according to their preferences, causing discomfort for those who want to adjust lighting or temperature in their workspace Hirning, Isoardi [53].

Our study found that green buildings were perceived as more comfortable and supportive of occupant productivity than non-certified buildings, but the difference was negligible for comfort and only small for perceived productivity. Interestingly, the results of this study indicated that the enablers of comfort and productivity are not the same. Our finding supports previous work by Rasheed et al. [28], who observed that the factors that influence how comfortable occupants are do not necessarily contribute to their productivity. Our study showed that although there were no significant differences in overall comfort among occupants, perceived productivity was only moderately better in green buildings.

Air quality, noise from colleagues, noise from outside, natural lighting, design, space in buildings, storage, work requirement, space at desks and personal control over heating and noise significantly influence perceived productivity in both green and non-certified buildings. No differences in terms of the parameters that influence productivity was found when comparing green with non-certified buildings.

Similarly, some parameters influenced overall comfort in green and non-certified buildings including temperature in winter, needs, the building's image and sufficiency of furniture. The interesting observation was that noise from outside, storage availability, adequate space at desks, and personal control over heating affected occupant productivity, and not overall comfort. As Winston Churchill once said *we shape our buildings, and our buildings shape us*, we need to develop a better understanding of how buildings are affecting their occupants, regulating their mood, productivity, and comfort. Green certification systems might benefit from further emphasis and credentials related to environmental parameters, particularly acoustic performance, artificial lighting, user-friendly IEQ controls and encouraging the development of POE protocols.

We must acknowledge a common limitation of the perception study described in this paper, which creates potential biases. Perceived satisfaction scores are only an indication of and not the actual evidence of a building's performance. However, occupants remain the best measurement approach to investigate how buildings affect their comfort and productivity. We recommend physical measurements of building performance as complementary to perception studies for a robust POE of buildings.

## 6. Conclusions

Based on the analysis of a dataset featuring responses from 5098 occupants from 68 buildings, the strengths of green buildings were identified as:

- design, image of the building, and cleaning, which were classified under operational parameters in this study.

The weaknesses of green buildings were identified as:

- All environmental parameters particularly too much noise from colleagues and unsatisfactory artificial lighting;
- Little or no personal control over indoor environmental parameters.

Although green buildings performed slightly better than non-certified in most parameters the differences were only small for design, image of the building, and cleaning. Green buildings were scored worse than non-certified buildings in terms of noise from colleagues and artificial lighting.

The study also revealed some areas worthy of further analysis within green building studies by finding the most influential parameters in perceived productivity and perceived comfort in both green and non-certified buildings. For perceived comfort, air quality, design, and work requirement were the most influential parameters affecting comfort satisfaction scores. For perceived productivity, noise was the most influential parameter on regulating productivity scores for both building groups.

**Author Contributions:** M.K., E.R. and G.B. participated equally in the conception of the idea, development of the methodology, analysis and interpretation of data, drafting of the article, and critical reviews. All authors have read and agreed to the published version of the manuscript.

**Funding:** This research received no external funding.

**Institutional Review Board Statement:** The study was conducted in accordance with the Declaration of Helsinki, and approved by the Institutional Review Board (or Ethics Committee) of Griffith University.

**Informed Consent Statement:** Informed consent was obtained from all subjects involved in the study.

**Data Availability Statement:** Not applicable.

**Acknowledgments:** It is a pleasure to acknowledge Adrian Leaman of Building Use Studies for permission to use the BUS Methodology questionnaire under license. We must also thank all the building managers and occupants for their participation in the surveys.

**Conflicts of Interest:** Authors declare no conflict of interest.

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
