# Peer review of "Do Green Buildings Have Superior Performance over Non-Certified Buildings? Occupants’ Perceptions of Strengths and Weaknesses in Office Buildings"

_buildings, doi:10.3390/buildings12091302_

Round 1

Reviewer 1 Report

1. Table 2 has not been mentioned in the text.

2. Table 6 should reports F like Table 4 and Table 5.

3.There are 9 operational parameters in table 1, but 10 operational parameters have been reported in table 5, 9 and 10. That means 'space at desk' has not been included in surveys but represented in the results.

4. In table 9 and 10, i suggest that 'Sig.' can been shown like ***p<0.001, **p<0.01, *p<0.5 respectively.   

Author Response

  1. Table 2 has not been mentioned in the text.

Response: We corrected this typo and cited Table 2 in the text. See Line 198 as we added (A summary of building specifications is the dataset in presented in Table 2.)

  1. Table 6 should reports F like Table 4 and Table 5.

Response: We corrected this typo and included F for Table 6. See the revised manuscript.

3.There are 9 operational parameters in table 1, but 10 operational parameters have been reported in table 5, 9 and 10. That means 'space at desk' has not been included in surveys but represented in the results.

Response: There are TEN operational parameters in both surveys and results. We previously missed to list Space at desks as the last parameter in operational parameters. We corrected this typo. See the updated Table 1.

  1. In table 9 and 10, i suggest that 'Sig.' can been shown like ***p<0.001, **p<0.01, *p<0.5 respectively.   

Response: as the reviewer suggested we revised the reporting of Significance values and presented the significance of results with more details by providing three types of stars for three different boundaries as ***p<0.001, **p<0.01, *p<0.5, see the footnotes in Table 9 and 10.

Reviewer 2 Report

The manuscript “Do green buildings have superior performance over non-certified buildings? Occupants’ perceptions of strengths and weaknesses” by Khoshbakht, M., et. al. presents a well-structured paper with clear results and discussions. A minor general comment is that I found that the title may need be clearer on the type of buildings it addresses, as it is clear that the study focuses in a particular type of buildings. Some detailed comments:

L31-33 Please present a clear definition of green buildings

L189       It could also help the reader to define whether a scale is unipolar or bipolar in table 1

L513       Sentence incomplete? Work requirement…

Author Response

The manuscript “Do green buildings have superior performance over non-certified buildings? Occupants’ perceptions of strengths and weaknesses” by Khoshbakht, M., et. al. presents a well-structured paper with clear results and discussions. A minor general comment is that I found that the title may need be clearer on the type of buildings it addresses, as it is clear that the study focuses in a particular type of buildings. Some detailed comments:

Response: We included office buildings in the title to clarify the type of buildings in the study.

L31-33 Please present a clear definition of green buildings

Response: We added a sentence in Line 31 that (Green buildings are those structures that are environmentally responsible and resource efficient throughout their life cycle.), to make it clear what is a green building as we continued our exact definition of green buildings in this manuscript by saying that Green buildings are in this paper categorised as buildings that had certifications from green building organisations such as Green Globe, Green Star, BRE, LEED, Green Design, etc.

L189       It could also help the reader to define whether a scale is unipolar or bipolar in table 1

 Response: The surveys were in the form of 7-point scales. Some of them were bipolar and some were unipolar, see the updated Table 1 for this.  

L513       Sentence incomplete? Work requirement…

Response: Work requirement was a typo in L513 and we deleted those.

Reviewer 3 Report

1. More than half of the buildings in the dataset were in New Zealand.  New Zealand's weather conditions are different from those of the other countries included in the analysis. Environmental factors could have biased the results. Especially in buildings with natural ventilation. It should be explained how the authors took this fact in results.

2. Would the results be similar if the amount of New Zealand buildings was less than 57%?

3. The authors reported that the buildings were less than 50 years old. The age of the building affects the efficiency of the systems (e.g. ventilation and air conditioning). More detailed information on the analyzed buildings should be provided.

4. What "Group 1" and "Group 2" mean in table 4?

Author Response

1. 

The subject of the presented paper is relevant for whoever wants to design a good quality building that perform in such a way that its users feel comfortable and productive.

However, the way the study is presented shows important weaknesses in its conception and its results fail to be useful for designers and for the building construction sector.

If the data presented are all the authors have about the buildings they have studied then I would say that this is not a paper to be published in “Buildings”. However, if there is more information that the authors can give about the studied buildings and if they rethink the way they present the study, maybe the paper can be reconsidered.

Response: Response: We appreciate the fact that the reviewer raised this discussion and pointed to the fact that this manuscript examined the weaknesses of green buildings. However, as the authors were designing the concept of this research, what they had in mind and aimed for was to provide a critique of green buildings and identify the areas in which green buildings are not necessarily performing better than non-certified buildings given the higher construction costs of green buildings. This is an evidence-based research approach and not a design paper. We didn’t aim particularly to suggest solutions for designers how to improve those weaknesses but rather identifying current issues with green buildings. We disagree with the reviewers’ comments about the significance of the results, and believe this manuscript is definitely useful for designer and practitioners by identifying the areas that need advancing in technology or design development in the future to design and construct better green buildings. 

The reader needs much more information about the studied buildings. The comparisons between green-certified building and non-certified buildings cannot be done like this. It seems that the authors forget that certification is not mandatory, a building may be non-certified and present a high environmental standard anyway. The studied building sample must be characterised in architectural and construction terms, in order to prove that the comparison made is suitable. How similar were the green-certified and non-certified buildings? Were there factors like location, orientation, age, etc. that may influence the results? Green-certification in itself is certainly not the main factor for the differences found between the two groups of buildings!

Response: We needed a criteria to classify buildings in our dataset into two groups of green and non-green buildings and the best criteria available to us (as used in many papers in the past) was green certification. However, we agree with the reviewer that some of the new buildings may have environmentally friendly features, but green certification is more likely to influence perceptions of occupants (as general public) than those environmentally friendly features that are more noticed to architects and designers in the field.   

Along the uploaded article pdf, the authors can find a number of other comments.

Response: We addressed all comments that were highlighted in the attached pdf file. See the revised manuscript.

In Text:

The comment about the title: In the title, we mentioned that we are studying occupants perceptions and readers should know it is a subjective study. The title is Do green buildings have superior performance over non-certified buildings? Occupants’ perceptions of strengths and weaknesses in office buildings.

This is a study which compares the green with non-certified buildings from occupant perspectives and identifies in which areas and aspects according to BUS Methodology green buildings still don’t show a superior performance over non-certified buildings. 

As listed in the conclusion, strengths of green buildings are:

  • design, image of the building, and cleaning, which were classified under operational parameters in this study.

AND weaknesses are:

  • All environmental parameters particularly too much noise from colleagues and unsatisfactory artificial lighting.
  • Little or no personal control over indoor environmental parameters.

 The comment in the abstract: There is a more detailed description about the buildings in our sample in Section 3.2, data characteristics, where we provide some of the most important buildings specifications in both groups of green and non-certified buildings. And as mentioned above our aim was to have a global review of green buildings given the 37 green buildings and 31 non-certified buildings in our dataset. This is not a controlled sample that all 38 buildings be identical in all aspects to 30 non-certified building group. Those kind of studies with controlled sample groups are normally done for 1 or 2 selective buildings where researchers select their case studies based on the similarities for the comparison reasons. But for a sample as big as 68 buildings, reviewers should know that finding controlled sample study for 68 buildings is very difficult if not impossible.

Definition of ΔM: We added a sentence explaining what ΔM is. ΔM is the median score of green buildings minus the median score of non-certified buildings.

Content of Table 4: We added some foot notes and explained what groups, N, F and sig means. See the revised manuscript.

The last comment in the Pdf file: This study is a POE of survey results investigating occupants' perspectives about green buildings, so this is a subjective study. We acknowledge this in the final concluding sentences to make clear to readers our investigation tools, but it doesn't make the results of this paper less significant as  we mentioned in the paragraph, occupants are the best instruments for measuring how well buildings are performing - even better than a thermometer or a anemometer. If occupants feel it is too cold in the building or it's too hot, we don't need a thermometer to confirm this. It is hot or it is cold. At the end of the day, it is occupants who use buildings and what they think matters, no matter what number my thermometer is showing.

Further comments from Reviewer 4 in the reviewer Pdf file are addressed and responses are added to the file.

Reviewer 4 Report

The subject of the presented paper is relevant for whoever wants to design a good quality building that perform in such a way that its users feel comfortable and productive.

However, the way the study is presented shows important weaknesses in its conception and its results fail to be useful for designers and for the building construction sector.

If the data presented are all the authors have about the buildings they have studied then I would say that this is not a paper to be published in “Buildings”. However, if there is more information that the authors can give about the studied buildings and if they rethink the way they present the study, maybe the paper can be reconsidered.

The reader needs much more information about the studied buildings. The comparisons between green-certified building and non-certified buildings cannot be done like this. It seems that the authors forget that certification is not mandatory, a building may be non-certified and present a high environmental standard anyway. The studied building sample must be characterised in architectural and construction terms, in order to prove that the comparison made is suitable. How similar were the green-certified and non-certified buildings? Were there factors like location, orientation, age, etc. that may influence the results? Green-certification in itself is certainly not the main factor for the differences found between the two groups of buildings!

Along the uploaded article pdf, the authors can find a number of other comments.

Author Response

The subject of the presented paper is relevant for whoever wants to design a good quality building that perform in such a way that its users feel comfortable and productive.

However, the way the study is presented shows important weaknesses in its conception and its results fail to be useful for designers and for the building construction sector.

If the data presented are all the authors have about the buildings they have studied then I would say that this is not a paper to be published in “Buildings”. However, if there is more information that the authors can give about the studied buildings and if they rethink the way they present the study, maybe the paper can be reconsidered.

Response: Response: We appreciate the fact that the reviewer raised this discussion and pointed to the fact that this manuscript examined the weaknesses of green buildings. However, as the authors were designing the concept of this research, what they had in mind and aimed for was to provide a critique of green buildings and identify the areas in which green buildings are not necessarily performing better than non-certified buildings given the higher construction costs of green buildings. This is an evidence-based research approach and not a design paper. We didn’t aim particularly to suggest solutions for designers how to improve those weaknesses but rather identifying current issues with green buildings. We disagree with the reviewers’ comments about the significance of the results, and believe this manuscript is definitely useful for designer and practitioners by identifying the areas that need advancing in technology or design development in the future to design and construct better green buildings. 

The reader needs much more information about the studied buildings. The comparisons between green-certified building and non-certified buildings cannot be done like this. It seems that the authors forget that certification is not mandatory, a building may be non-certified and present a high environmental standard anyway. The studied building sample must be characterised in architectural and construction terms, in order to prove that the comparison made is suitable. How similar were the green-certified and non-certified buildings? Were there factors like location, orientation, age, etc. that may influence the results? Green-certification in itself is certainly not the main factor for the differences found between the two groups of buildings!

Response: We needed a criteria to classify buildings in our dataset into two groups of green and non-green buildings and the best criteria available to us (as used in many papers in the past) was green certification. However, we agree with the reviewer that some of the new buildings may have environmentally friendly features, but green certification is more likely to influence perceptions of occupants (as general public) than those environmentally friendly features that are more noticed to architects and designers in the field.   

Along the uploaded article pdf, the authors can find a number of other comments.

Response: We addressed all comments that were highlighted in the attached pdf file. See the revised manuscript.

In Text:

The comment about the title: In the title, we mentioned that we are studying occupants perceptions and readers should know it is a subjective study. The title is Do green buildings have superior performance over non-certified buildings? Occupants’ perceptions of strengths and weaknesses in office buildings.

This is a study which compares the green with non-certified buildings from occupant perspectives and identifies in which areas and aspects according to BUS Methodology green buildings still don’t show a superior performance over non-certified buildings. 

As listed in the conclusion, strengths of green buildings are:

  • design, image of the building, and cleaning, which were classified under operational parameters in this study.

AND weaknesses are:

  • All environmental parameters particularly too much noise from colleagues and unsatisfactory artificial lighting.
  • Little or no personal control over indoor environmental parameters.

 The comment in the abstract: There is a more detailed description about the buildings in our sample in Section 3.2, data characteristics, where we provide some of the most important buildings specifications in both groups of green and non-certified buildings. And as mentioned above our aim was to have a global review of green buildings given the 37 green buildings and 31 non-certified buildings in our dataset. This is not a controlled sample that all 38 buildings be identical in all aspects to 30 non-certified building group. Those kind of studies with controlled sample groups are normally done for 1 or 2 selective buildings where researchers select their case studies based on the similarities for the comparison reasons. But for a sample as big as 68 buildings, reviewers should know that finding controlled sample study for 68 buildings is very difficult if not impossible.

Definition of ΔM: We added a sentence explaining what ΔM is. ΔM is the median score of green buildings minus the median score of non-certified buildings.

Content of Table 4: We added some foot notes and explained what groups, N, F and sig means. See the revised manuscript.

The last comment in the Pdf file: This study is a POE of survey results investigating occupants' perspectives about green buildings, so this is a subjective study. We acknowledge this in the final concluding sentences to make clear to readers our investigation tools, but it doesn't make the results of this paper less significant as  we mentioned in the paragraph, occupants are the best instruments for measuring how well buildings are performing - even better than a thermometer or a anemometer. If occupants feel it is too cold in the building or it's too hot, we don't need a thermometer to confirm this. It is hot or it is cold. At the end of the day, it is occupants who use buildings and what they think matters, no matter what number my thermometer is showing.

Further comments from Reviewer 4 in the reviewer Pdf file are addressed and responses are added to the file.

Round 2

Reviewer 4 Report

The authors did not address satisfactorily all the questions raised.

I still disagree with the kind of approach of the study. This is more a satisfaction inquiry than a construction science study. It can be useful for promoters but it fails to be useful for researchers in the field. In my opinion, this does not fit into Buildings scope. However, the other reviewers did not object to it, so I may be wrong.

The most important is, in my opinion, the fact that the lack of information about the sample of buildings may lead to erroneous conclusions. As said before, the differences in the occupants’ perception may be related to other features of the buildings and not to the fact that they are non-certified or green buildings.

Of course a sample of 68 buildings is not a controlled sample, but some criteria must have been followed for the selection of the buildings in this sample. Furthermore, a comparative study was meant, so the authors certainly chose comparable buildings. However, not knowing the main construction characteristics of the buildings, some of them impacting greatly in occupants’ comfort and well-being, make the readers unable of judging the results. Therefore, at least some more information about the buildings should be given.

Author Response

I still disagree with the kind of approach of the study. This is more a satisfaction inquiry than a construction science study. It can be useful for promoters but it fails to be useful for researchers in the field. In my opinion, this does not fit into Buildings scope. However, the other reviewers did not object to it, so I may be wrong.

Response: This type of studies are called subjective studies and they are vastly common in the world of building performance assessment. These type of studies of occupant satisfaction analysis are useful for building diagnostics and operations. Please see the following papers from well-known authors as a proof of credibility of occupants satisfaction studies.

  1. Candido, C., Kim, J., de Dear, R., & Thomas, L. (2016). BOSSA: A multidimensional post-occupancy evaluation tool. Building Research & Information44(2), 214-228.
  2. Kent, M., Parkinson, T., Kim, J., & Schiavon, S. (2021). A data-driven analysis of occupant workspace dissatisfaction. Building and Environment205, 108270.
  3. Cheung, T., Schiavon, S., Graham, L. T., & Tham, K. W. (2021). Occupant satisfaction with the indoor environment in seven commercial buildings in Singapore. Building and Environment188, 107443.

These are only a few but there is a great body of research in building pefrmance studies that rely solely on occupant satisfaction.

The most important is, in my opinion, the fact that the lack of information about the sample of buildings may lead to erroneous conclusions. As said before, the differences in the occupants’ perception may be related to other features of the buildings and not to the fact that they are non-certified or green buildings.

Response: Please see the following paragraphs from Line 128 and Line 211 the information about the sample of buildings and how we carefully selected our 68 buildings to make the results representable. Please also see Tables 2 and 3 that are exclusively showing characteristics of the buildings in our dataset.

Line 128: Our study analyses the POEs of 68 buildings globally comprising green and non-certified buildings. In the building sample selection, a few criteria were advanced to keep the dataset consistent and make sure results are representative. The first criterion was that the selected building for study should have had offices for staff that spend most of their time indoors. The second criterion was the building type, making sure the selected building was either commercial office building or university building. We also made sure we could collect responses from at least 75% of the occupants and that the buildings were less than 50 years old. Our sample of green buildings included various global green building certifications including Green Globe, Green Star, BRE, LEED, and Green Design.

All the buildings had both cellular offices and open-plan areas shared by various occupants. The buildings include commercial and academic offices accommodating meeting spaces, seminar rooms, and offices for staff. All green buildings feature a mixed-mode ventilation system. Mixed-mode buildings are those which use a combination of air-conditioning and natural ventilation through windows and air vents. The control of mixed-mode ventilations and switching between air-conditioning and natural ventilation differ among our sample of green buildings. In some green buildings, the control of the mixed-mode ventilation system is fully automated with no opportunity for occupants to override the automated operation of windows (see Figure 1a). In this type, the ventilation system switches between air-conditioning and natural ventilation only if the outdoor temperature falls within the desired temperature range, and occupants have no control over the ventilation operation modes. In type 2 green buildings, occupants have the opportunity to over-ride the automated window operations and manually switch ventilation modes between air-conditioning and natural ventilation (see Figure 1b). In the rest of the building sample (type 3 green buildings), the decision over the mode of ventilation operation is made by occupants, and the automation system only prevents the concurrent operation of air-conditioning and natural ventilation by switching off the air-conditioning when windows are opened (see Figure 1c). In non-certified buildings, ventilation modes were either or both air-conditioning and natural ventilation, and occupants had full control of air-conditioning and operation of windows, the difference from green buildings being that there was no automation system to prevent the concurrent operation of the two ventilation modes, which may result in the loss of conditioned air through open windows and increasing building energy loads.

Line 211: A summary of building specifications in the dataset is presented in Table 2. As indicated in Table 2, just over half the buildings in our dataset were located in New Zealand (39 buildings), while others were in a range of countries, including Australia, England, USA, India, Ireland, Japan, Malaysia, and Canada. The average number of participants from each building in our dataset was 75. In terms of building use, our dataset consists of commercial buildings (72%), and academic buildings (28%). Commercial buildings are used primarily for business purposes, and academic buildings are used for university teaching, research and administration. For ventilation, 43% of buildings in our dataset had natural ventilation, 31% had air-conditioning, and 26% had a combination of natural ventilation and air-conditioning (mixed-mode) systems.

For both sets of buildings the majority of participants were 30 years or over (69% in green and 81% in non-certified buildings).  In green buildings the percentage rate of female participants was 54% and male participants was 46%, while in non-certified buildings the percentage rate of female participants was 47% and male participants was 53%. For both building type, almost half of the participants were seating next to a window (53% in green and 49% in non-certified buildings). Workgroups consist of five categories: solo with a single occupant, duo with two occupants, 3-4 with three or four occupants, 5-9 with five to nine occupants, and over 9 with more than nine occupants. In terms of space sharing with other, most participants in both building groups the room was shared with over 9 people. More details of spaces sharing in comparing the two building types are provided in Table 3 for comparison purposes.

Of course a sample of 68 buildings is not a controlled sample, but some criteria must have been followed for the selection of the buildings in this sample. Furthermore, a comparative study was meant, so the authors certainly chose comparable buildings. However, not knowing the main construction characteristics of the buildings, some of them impacting greatly in occupants’ comfort and well-being, make the readers unable of judging the results. Therefore, at least some more information about the buildings should be given.

Response: As explained here earlier we had a selection criteria and we explained our building selection process in Line 128 as typed above.

….. In the building sample selection, a few criteria were advanced to keep the dataset consistent and make sure results are representative. The first criterion was that the selected building for study should have had offices for staff that spend most of their time indoors. The second criterion was the building type, making sure the selected building was either commercial office building or university building. We also made sure we could collect responses from at least 75% of the occupants and that the buildings were less than 50 years old. Our sample of green buildings included various global green building certifications including Green Globe, Green Star, BRE, LEED, and Green Design…..
